# The rules of multiplayer cooperation in networks of communities

**Diogo L. Pires** *, **Mark Broom**

Department of Mathematics, City, University of London, Northampton Square, London, United Kingdom

* Diogo.L.Pires@city.ac.uk

## Abstract

Community organisation permeates both social and biological complex systems. To study its interplay with behaviour emergence, we model mobile structured populations with multiplayer interactions. We derive general analytical methods for evolutionary dynamics under high home fidelity when populations self-organise into networks of asymptotically isolated communities. In this limit, community organisation dominates over the network structure and emerging behaviour is independent of network topology. We obtain the rules of multiplayer cooperation in networks of communities for different types of social dilemmas. The success of cooperation is a result of the benefits shared among communal cooperators outperforming the benefits reaped by defectors in mixed communities. Under weak selection, cooperation can evolve and be stable for any size ($Q$) and number ($M$) of communities if the reward-to-cost ratio ($V/K$) of public goods is higher than a critical value. Community organisation is a solid mechanism for sustaining the evolution of cooperation under public goods dilemmas, particularly when populations are organised into a higher number of smaller communities. Contrary to public goods dilemmas relating to production, the multiplayer Hawk-Dove (HD) dilemma is a commons dilemma focusing on the fair consumption of preexisting resources. This game yields mixed results but tends to favour cooperation under larger communities, highlighting that the two types of social dilemmas might lead to solid differences in the behaviour adopted under community structure.

**Data Availability Statement:** All relevant data are within the manuscript and its Supporting information files.

**Funding:** DLP and MB received funding from the European Union's Horizon 2020 research and

## Author summary

Human and animal behaviour is strongly influenced by the structure of their social interactions. The interaction networks that characterise these social systems are diverse, but among them community structure can be often found. This work focuses on the impact of community organisation on the evolution of cooperative behaviour in the face of collective social dilemmas. The work shows that community organisation is a solid mechanism for sustaining the evolution of cooperation under public goods dilemmas, particularly when populations are organised into a larger number of smaller communities. However, cooperation can evolve for any size and number of communities under all public goods dilemmas considered. The success of cooperation is a result of the benefits shared among communal cooperators outperforming the benefits reaped by defectors in mixed

innovation programme under the Marie Skłodowska-Curie grant agreement No 955708 (https://rea.ec.europa.eu/funding-and-grants/horizon-europe-marie-sklodowska-curie-actions_en). The funders had no role in study design, data collection and analysis, decision to publish, or preparation of the manuscript.

**Competing interests:** The authors have declared that no competing interests exist.

communities. The size and number of communities is much more important in determining the evolution of cooperation than the way they are connected to other communities. These results have significant implications for the study of animal and human social behaviour, metapopulation dynamics, and general dynamics on social interaction networks.

## 1 Introduction

Understanding how individuals organise into social communities is of interest to various research fields due their ubiquitous presence in social systems. This is shown by the study of networks of friendships, academic collaborations, individual interests, online discourse, and political affiliation, among other social interaction systems [1–4]. Its organisation occurs down to the smallest scale of human societies, which has motivated looking at the small interaction groups in which we partake as a core configuration of our social psychology [5]. This has been further supported by experimental studies showing that small groups, and their limit of dyadic interactions, constitute most of our social encounters [6]. Animal groups often organise themselves into social communities as well [7]. Their formation can be motivated by the fragmentation of habitats, and its subsequent impact on ecological networks has led to the study of evolution in metapopulations [8, 9]. Even in the presence of migration fluxes involving roaming great distances, animals may maintain the same community and social ties, either by collectively coordinating their movements [10, 11], or by coming back to the same territorial patches where they once settled [12–14].

The organisation of individuals into social communities significantly influences their behaviour with one another, particularly when facing social dilemmas. Social dilemmas embody the conflict between social and individual interests, often framed as a choice one has to make between cooperating and defecting, the dynamics of which have been extensively modelled using evolutionary game theory. Incorporating community structure into these models has thus far entailed considering events of two different natures: within-community reproduction and between-community migration. These models are typically referred to as metapopulation dynamics, a classification of which has been performed in [15]. The distinct nature of between-community events has been further emphasised by considering community-level events, such as group reproduction [16] or group splitting [17, 18], which involve the replacement of entire groups either by other groups or by single individuals. Others have considered different intensities of selection acting on within- and between-community events [19, 20]. Some of these modelling features suggest inspiration from multilevel selection to different degrees, which we intentionally avoid in our current work. Although these approaches lead to the evolution of pairwise cooperation, they may rely on the distinct nature of between-community events to do so, or even on additional mechanisms present such as punishment strategies [19].

Furthermore, metapopulation models generally assume that communities are connected to each other in the same way, with few exceptions to this [16] as is pointed out in [21]. However, other features of social interaction networks have been shown to have a strong interplay with the evolution of cooperation in pairwise dilemmas. These include low average degree [22], small-world characteristics [23], high link heterogeneity [24], and strong pair ties [25]. Some of these effects may be sensitive to the evolutionary dynamics considered [22, 26], although the qualitative differences have been shown to vanish under a generalisation of the dynamics [27]. The extension of these population structure models to multiplayer interactions is not trivial

and considering only lower-order networks with dyadic interactions is often insufficient to represent them [28]. Here, we will focus on one model of multiplayer interactions where both network and community structure are conveniently integrated.

The framework introduced in its general form in [29] offers a novel approach to multiplayer social dilemmas, where interacting groups of individuals emerge from their simultaneous presence on the nodes of a spatial network. The model operates under the minimal assumption that typical evolutionary dynamics on graphs, such as birth-death, death-birth or link dynamics [26, 30], act between any two individuals in the population depending on their frequency of interaction within the same group. Various movement models have been explored so far, an overview of which is provided in [31]. Movement contingent on satisfaction with past interactions sustains the co-evolution of cooperation and assortative behaviour, especially under complete networks [32, 33] and for several evolutionary dynamics [34]. Mobility costs are essential to determine whether cooperative behaviour emerges [34], parallel to what is reported from more realistic spatial social dilemmas [35]. Alternatively, the territorial raider model in the form introduced later in this paper has been used to study the fully independent movement of individuals around their home nodes, governed by one single parameter, the individuals' home fidelity. This model has revealed more limited prospects for the evolution of cooperation within small networks [36], small fully connected communities [26], and intermediate-sized complex networks with diverse structural properties [37, 38].

We propose the use of this fully independent movement model to study evolutionary dynamics in network- and community-structured populations with multiplayer interactions. Our focus centres on the limit of high home fidelity, where communities exhibit asymptotically low mobility. In section 3.1, we derive general analytical methods for the dynamics in this limit. We conclude that the organisation of the population into a network of communities uniquely influences the evolutionary dynamics through the number and size of the communities, rather than through the way communities are connected. Some dynamics amplify within-community selection and others increment between-community events. In section 3.2, we show that the balance between the two types of events determines whether cooperation evolves, and we obtain their contributions to fixation probabilities under weak selection for several social dilemmas. In section 3.3, we use this balance to derive the rules of multiplayer cooperation and compare them among social dilemmas. In section 3.4, we analyse in detail one particular game, the Charitable Prisoner's Dilemma, and draw a comparison with some of the results obtained in the widely explored pairwise donation game. In section 4, we connect our findings to the relevant literature on multiplayer social dilemmas, metapopulation dynamics, and mobile structured populations. Once again showcasing its versatility, this framework enabled the exploration of network and community structure, thereby revealing the high potential for the evolution of cooperation across diverse social dilemmas.

## 2 The model

The general framework introduced in [29] has been used to study the interplay between population structure, movement and multiplayer interactions. Here, we focus on the territorial raider model, a model of fully independent movement, which was generalised in [26] to account for subpopulations or, as we will refer to them, communities. We start by defining structure and the movement rules of this model. We then revisit the general approach to social dilemmas outlined in [39], and finish by presenting the set of evolutionary dynamics defined in [26].

## 2.1 Population structure and movement

A population is composed of $N$ individuals $I_n = I_1, \ldots, I_N$. Individuals are positioned on a spatial network with $M$ places $P_m = P_1, \ldots, P_M$, which has a set of edges connecting them. Even though the terms graph and network are often used interchangeably in the literature, here we follow the same terminology used in [34, 37]. The term graph will only be used for the underlying evolutionary graph representing the replacement structure between **individuals**, and network will be used to refer to the network of **places**.

Under fully independent movement models, the position of each individual is independent both of where they were previously and of where other individuals will be [29]. Therefore, the probability that an individual $I_n$ is in place $P_m$ is generally defined by $p_{nm}$. Under the territorial raider model, each individual has an assigned home node in the network, and the probability distribution of their positions is defined as the following:

$$
p_{nm} = \begin{cases} h/(h + d_n), & n = m, \\ 1/(h + d_n), & n \neq m \text{ and vertices } n, m \text{ connected}, \\ 0, & \text{otherwise}, \end{cases} \tag{1}
$$

where $h$ is the home fidelity parameter, and $d_n$ is the degree of the home node of individual $I_n$. This movement model is governed by a single parameter $h$ yet allows for different movement propensities governed by the opportunities available to each individual, reflecting basic characteristics of local limited mobility present in animal populations based on territorial behaviour [12–14] as well as human social systems. Alternative models could be used, some of which would lead to exactly the same results, as it is briefly discussed in the next section. We use the version of the territorial raider model under which each node of the network is home to a community of $Q$ individuals, and thus $M$ denotes the number of communities and $N = MQ$. The probability distribution of positions under the territorial raider model is represented in Fig 1. Communities have been referred to in previous models as subpopulations [26] or demes [20]. The below definitions are valid for any distribution $p_{nm}$ of a fully independent movement model.

A group of individuals $\mathcal{G}$ has probability $\chi(m, \mathcal{G})$ of meeting in node $P_m$, which is given by:

$$
\chi(m, \mathcal{G}) = \prod_{i \in \mathcal{G}} p_{im} \prod_{j \notin \mathcal{G}} (1 - p_{jm}). \tag{2}
$$

The fitness of each individual $I_n$ is obtained through the weighted average of the payoffs $R_{n,m,\mathcal{G}}$ received in each place $P_m$ and each group composition $\mathcal{G}$ they can be in. We further introduce $w$, the intensity of selection as defined in [40], which measures the extension to which the outcomes of the game contribute to the fitness of individuals:

$$
F_n = 1 - w + w \sum_m \sum_{\mathcal{G}:n \in \mathcal{G}} \chi(m, \mathcal{G}) R_{n,m,\mathcal{G}}. \tag{3}
$$

We bring attention to an alternative notation used in the literature, where a background payoff defined as $R$ is introduced. This notation is used under movement models such as those from [26, 33, 34, 36]. The background payoff is typically included within the effective reward received in each interaction, which we denote $R'_{n,m,\mathcal{G}} = R_{n,m,\mathcal{G}} + R$. This leads to the following adjustments to the fitness of individuals:

$$
F'_n = \sum_m \sum_{\mathcal{G}:n \in \mathcal{G}} \chi(m, \mathcal{G}) R'_{n,m,\mathcal{G}}. \tag{4}
$$

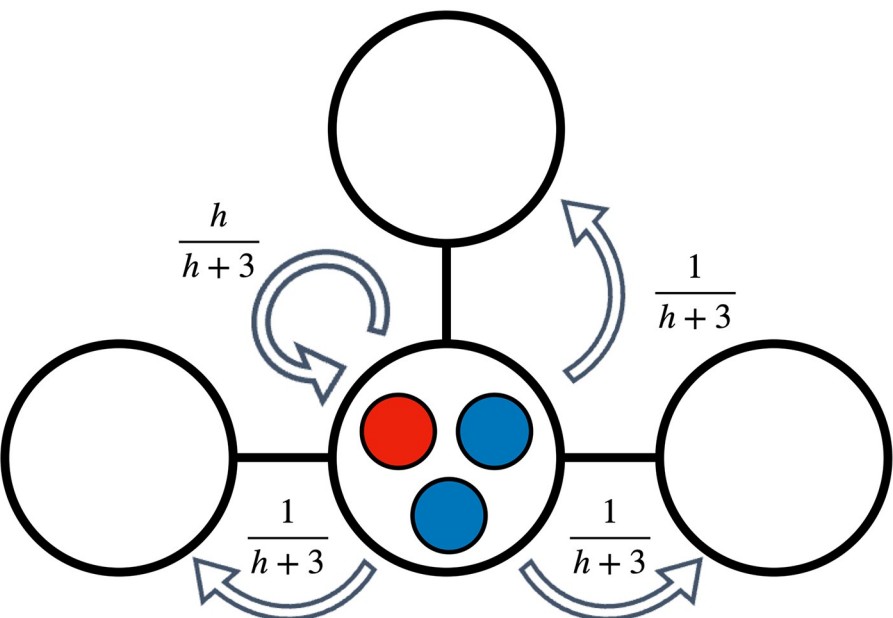

**Fig 1. Representation of a small community network under the territorial raider model.** At each time step, individuals initiate their movement from their home node, and can either remain there or move to any of the adjacent nodes, returning to their home node prior to the next time step. In this figure, we represent the resulting probability distribution for the community in the centre of the network.

We will make use of the first notation where the intensity of selection is used, as this is revealed to be more practical when inspecting the weak selection limit. Nonetheless, the second approach leads to a simple rescaling of the fitness $F' = \frac{1}{w}F$ when $R = \frac{1-w}{w}$, which has no impact on the evolutionary dynamics introduced later.

## 2.2 Multiplayer social dilemmas

We consider the multiplayer social dilemmas studied in [39]. Individuals have two strategies available to them: to cooperate (C) or to defect (D). In these dilemmas, payoffs can be represented as $R_{n,m,\mathcal{G}} \equiv R_{c,d}^C (\equiv R_{c,d}^D)$ when the focal individual $I_n$ is a cooperator (defector), as they are determined by the type of the focal individual and the number of cooperators $c$, and defectors $d$ in their current group. We present the payoffs received under each social dilemma in Table 1, where $V$ represents the value of the reward shared, and $K$ the cost paid by individuals in the group. In public goods dilemmas, cooperation involves the production of a reward $V$ at a cost $K$, which is consumed by individuals within the group. In contrast, commons dilemmas typically represent scenarios with preexisting resources, where cooperation can involve, among other things, the sustainable consumption of the resources. In the HD dilemma, the only commons dilemma we study here, cooperators evenly share the reward $V$, while defectors attempt to consume it entirely, either winning it occasionally or losing it to other defectors while incurring a cost $K$.

## 2.3 Evolutionary dynamics

We follow an approach grounded on evolutionary graph theory [46]. The population has a corresponding evolutionary graph represented by the adjacency matrix $\mathbf{W} = (w_{ij})$, with $w_{ij}$

**Table 1. Payoffs obtained by a focal cooperator $R_{c,d}^C$ or a focal defector $R_{c,d}^D$ in a group with $c$ cooperators and $d$ defectors playing a social dilemma.** Social dilemmas are referred to in the text by the acronyms introduced in this table.

| Multiplayer Game | $R_{c,d}^C$ | $R_{c,d}^D$ |
|---|---|---|
| Charitable Prisoner's Dilemma (CPD) [36] | $\begin{cases} \dfrac{c-1}{c+d-1}V - K & c > 1 \\ -K & c = 1 \end{cases}$ | $\begin{cases} \dfrac{c}{c+d-1}V & c > 0 \\ 0 & c = 0 \end{cases}$ |
| Prisoner's Dilemma (PD) [41] | $\dfrac{c}{c+d}V - K$ | $\dfrac{c}{c+d}V$ |
| Prisoner's Dilemma with Variable production function (PDV) [42] | $\dfrac{V}{c+d}\sum_{n=0}^{c-1}\omega^n - K, w > 0$ | $\dfrac{V}{c+d}\sum_{n=0}^{c-1}\omega^n, w > 0$ |
| Volunteer's Dilemma (VD) [43] | $V - K$ | $\begin{cases} V & c > 0 \\ 0 & c = 0 \end{cases}$ |
| Snowdrift (S) [42] | $V - \dfrac{K}{c}$ | $\begin{cases} V & c > 0 \\ 0 & c = 0 \end{cases}$ |
| Threshold Volunteer's Dilemma (TVD) [42] | $\begin{cases} V - K & c \geq L \\ -K & c < L \end{cases}$ | $\begin{cases} V & c \geq L \\ 0 & c < L \end{cases}$ |
| Stag Hunt (SH) [44] | $\begin{cases} \dfrac{c}{c+d}V - K & c \geq L \\ -K & c < L \end{cases}$ | $\begin{cases} \dfrac{c}{c+d}V & c \geq L \\ 0 & c < L \end{cases}$ |
| Fixed Stag Hunt (FSH) [44] | $\begin{cases} \dfrac{V}{c+d} - K & c \geq L \\ -K & c < L \end{cases}$ | $\begin{cases} \dfrac{V}{c+d} & c \geq L \\ 0 & c < L \end{cases}$ |
| Threshold Snowdrift (TS) [45] | $\begin{cases} V - \dfrac{K}{c} & c \geq L \\ -\dfrac{K}{L} & c < L \end{cases}$ | $\begin{cases} V & c \geq L \\ 0 & c < L \end{cases}$ |
| Hawk–Dove (HD) [29] | $\begin{cases} \dfrac{V}{c} & d = 0 \\ 0 & d > 0 \end{cases}$ | $\dfrac{V - (d-1)K}{d}$ |

denoting the replacement weights which determine the likelihood of individual $I_i$ replacing $I_j$ in an evolutionary step. In contrast with the original formulation of evolutionary pairwise games on graphs, the interaction structure between individuals is an emerging feature of the model. We follow the procedure used in [26], under which replacement weights are determined by the fraction of time any two individuals spend interacting within the network. They spend their time equally with each of the other individuals in their groups, and time spent alone contributes to their self-replacement weights. This leads to the following definition:

$$w_{ij} = \begin{cases} \displaystyle\sum_m \sum_{\mathcal{G}:i,j\in\mathcal{G}} \frac{\chi(m,\mathcal{G})}{|\mathcal{G}|-1}, & i \neq j, \\ \displaystyle\sum_m \chi(m,\{i\}), & i = j. \end{cases} \tag{5}$$

Let us consider that the population goes through an evolutionary process operating on the strategies C and D used by each individual. This is modelled in discrete evolutionary steps, during which individuals may update their strategies. The probability that, at a given step, the strategy of an individual $I_i$ replaces that of $I_j$ is denoted by the replacement probability $\tau_{ij}$. This probability may depend in different ways on the fitness of individuals, thereby incorporating selection into the process, and on the replacement weights, thereby capturing their interaction structure. We recall the dynamics outlined in [26], and their respective replacement

**Table 2. Definition of birth probabilities ($b_{i(j)}$) and death probabilities ($d_{(i)j}$), or of final replacement probability ($\tau_{ij}$), for six distinct evolutionary dynamics.** The indices denote the individuals $I_i$ giving birth and $I_j$ dying. In instances where the replacement probability is not explicitly stated, it can be derived by multiplying the respective birth and death probabilities.

| Evolutionary dynamics and replacement probabilities | | | |
|---|---|---|---|
| BDB | $b_i = \dfrac{F_i}{\sum_n F_n}, d_{ij} = \dfrac{w_{ij}}{\sum_n w_{in}}$ | DBD | $d_j = \dfrac{F_j^{-1}}{\sum_n F_n^{-1}}, b_{ij} = \dfrac{w_{ij}}{\sum_n w_{nj}}$ |
| DBB | $d_j = 1/N, b_{ij} = \dfrac{w_{ij} F_i}{\sum_n w_{nj} F_n}$ | BDD | $b_i = 1/N, d_{ij} = \dfrac{w_{ij} F_j^{-1}}{\sum_n w_{in} F_n^{-1}}$ |
| LB | $\tau_{ij} = \dfrac{w_{ij} F_i}{\sum_{n,k} w_{nk} F_n}$ | LD | $\tau_{ij} = \dfrac{w_{ij} F_j^{-1}}{\sum_{n,k} w_{nk} F_k^{-1}}$ |

probabilities $\tau_{ij}$ are summarised in Table 2. The evolutionary dynamics are classified as birth-death (BD) if an individual is first selected for birth and then another one for death; death-birth (DB) if the reverse order of events is considered; and link (L) if an edge of the evolutionary graph is directly chosen. Under each of these, selection can act either on the birth (B) or the death (D) event. Evolutionary dynamics are thus referred to by the combination of these two codes, as is shown in Table 2.

We consider both the fitness and replacement weights of individuals to be computed assuming a large sample of random interactions within their environment, as has been widely done both in pairwise games [22–24], and multiplayer games [26, 34, 47]. Alternatively, these could have been calculated using different sampling assumptions, such as considering those two quantities to be obtained from two independent single interaction samples [37, 38]. In those cases, there might be other effects emerging if the sampling used to calculate both quantities is correlated, as was shown in [48].

The probability of fixation for a single mutant cooperator (defector) in a population with the opposing strategy is defined as $\rho^C$ ($\rho^D$). Selection is said to favour the fixation of cooperation if $\rho^C > \rho^{neutral}$, and it is said to favour its evolution if $\rho^C > \rho^{neutral} > \rho^D$ [40]. The neutral fixation probability is equal to $\rho^{neutral} = 1/N = 1/(MQ)$ [49]. Fixation probabilities can be calculated under the general fully independent movement models resorting to the proceeding explained in [26, 36]. However, in the results section, we concentrate on limits where fixation probabilities assume closed-form expressions.

## 3 Results

Let us consider the previously introduced model in the limit of high home fidelity $h \to \infty$. A description of the free parameters of the model and the limits considered is provided in section 1 of S1 File, particularly in Table A. In section 3.1, we describe the evolutionary process arising from this limit across the six introduced dynamics and derive exact expressions for single mutant fixation probabilities under any network of communities. The analysis in this section is substantiated by the work in section 2 of S1 File. In section 3.2, we analyse the expansion of fixation probabilities within the additional limit of weak selection, which unveils simple contributions of within-community fixation processes and between-community replacement events. We further analyse these contributions under the general social dilemma section. These findings are complemented by the content in section 3 of S1 File. In section 3.3, we present the simple rules obtained for the evolution of cooperation under the general multiplayer social dilemmas, when three successive limits are considered: high home fidelity, weak selection and large networks of communities. In section 4 of S1 File, we analyse the extent to which these rules are valid outside of the

limits of large networks and weak selection. Moreover, we contextualise the particular case of the CPD with respect to prior literature on pairwise dilemmas in section 3.4.

## 3.1 Evolutionary dynamics under high home fidelity

Consider a connected network comprising $M$ places and an arbitrary topology. Each place is home to a community of size $Q$ with movement following the territorial raider model (see Fig 1). In the asymptotic limit of high home fidelity $h \to \infty$, individuals interact mostly within their community. The fitness of each individual depends mainly on the rewards $R_{c,d}^C$ and $R_{c,d}^D$ received within each community of $c$ cooperators and $d$ defectors, higher-order terms on $h^{-1}$ dependent on the composition of the remaining communities. We define the asymptotic value of the fitness of a focal cooperator and defector as respectively the following:

$$f_{c,d}^C = 1 - w + wR_{c,d}^C, \tag{6}$$

$$f_{c,d}^D = 1 - w + wR_{c,d}^D. \tag{7}$$

In this limit, it is possible to obtain a closed-form expression for the fixation probability of a single mutant. The fixation process under each of the six introduced dynamics corresponds to a nested Moran process involving the fixation of a single mutant on its community and the fixation of that community in the population. A part of this process is represented in Fig 2. The probabilities obtained are presented in the next subsections (see section 2 of S1 File for more information about how they were obtained).

**3.1.1 Fixation probabilities under BDB, DBD, LB and LD dynamics.** In the context of high home fidelity, replacement events within the same community happen at an asymptotically larger rate than events between different communities. As such, fixation probabilities $\rho^C$ and $\rho^D$ are obtained by multiplying the probability of the original mutant fixating within its community, denoted as $r^C$ or $r^D$, by the probability of the community achieving fixation in the whole population. We note that these probabilities are identical under the BDB, DBD, LB and LD dynamics because the transition probability ratios that characterise the process are identical at any given state of the population.

Within-community fixation is equivalent to a frequency-dependent Moran process where the fitness of individuals corresponds to its asymptotic value in isolated communities as defined in Eqs 6 and 7. Fixation probabilities for cooperators and defectors are determined as follows:

$$r^C = \frac{1}{1 + \sum_{j=1}^{Q-1} \prod_{c=1}^{j} \frac{f_{c,Q-c}^D}{f_{c,Q-c}^C}}, \tag{8}$$

$$r^D = \frac{1}{1 + \sum_{j=1}^{Q-1} \prod_{d=1}^{j} \frac{f_{Q-d,d}^C}{f_{Q-d,d}^D}}. \tag{9}$$

Upon reaching a state with homogeneous communities, one of two state-changing events may unfold. In one scenario, a cooperator replaces a defector from an adjacent community, with probability proportional to its communal fitness $f_{Q,0}^C$. Subsequently, the new cooperator may fixate within that community with a probability of $r^C$. Alternatively, a defector may replace a cooperator from a different community, proportionally to $f_{0,Q}^D$, and the new defector may fixate within the new community with a probability of $r^D$. The fixation process of one

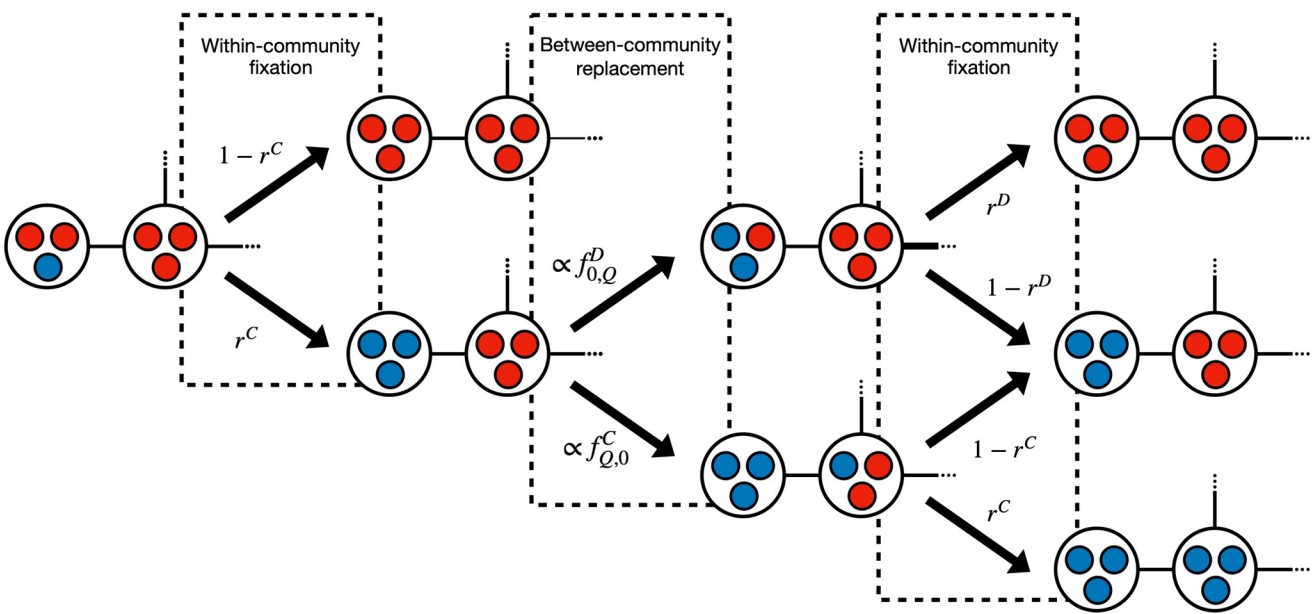

**Fig 2. Fixation process in a population of connected communities under the asymptotic limit of high home fidelity.** For simplicity, let us consider the scenario where one mutant cooperator emerges in a population of defectors. The new strategy will fixate within the community where it originated with a probability of $r^C$. The state attained has homogeneous communities and may change only through the occurrence of a between-community replacement. This involves either a cooperator replacing a defector from an adjacent community or the reverse, with probabilities proportional to their respective communal fitness $f_{Q,0}^C$ and $f_{0,Q}^D$. Each of those events may be followed by the within-community fixation of the new type, with respective probabilities $r^C$ and $r^D$. If within-community fixation is unsuccessful, it will result in the restoration of the previous number of homogeneous communities of cooperators. However, if within-community fixation is successful, it will respectively increase or decrease by one the number of communities of cooperators in the network. The transition probability ratio $\Gamma$ (see Eq 10) between these two possible state transitions is constant and can be obtained from this diagram. The represented probabilities are the same under BDB, DBD, LB and LD dynamics. Under DBB and BDD dynamics, within-community fixation probabilities are computed from Eqs 15 and 16, and the transition probability ratio is obtained from Eq 17.

community on the entire population is equivalent to a fixed fitness Moran process, where the transition probability ratio is as follows (see a visual representation in Fig 2 and a formal derivation in section 2.1 of S1 File):

$$\Gamma = \frac{f_{0,Q}^D \cdot r^D}{f_{Q,0}^C \cdot r^C}. \tag{10}$$

Please note that the ratio between the two within-community fixation probabilities can be considered in its following simplified form [40, 50]:

$$\frac{r^D}{r^C} = \prod_{c=1}^{Q-1} \frac{f_{c,Q-c}^D}{f_{c,Q-c}^C}. \tag{11}$$

The fixation probability of a single mutant cooperator or defector in a population of the opposing type is respectively the following:

$$\lim_{h \to \infty} \rho^C = r^C \cdot P_{Moran}\left(\Gamma^{-1}\right) = r^C \cdot \frac{1-\Gamma}{1-\Gamma^M}, \tag{12}$$

$$\lim_{h \to \infty} \rho^D = r^D \cdot P_{Moran}(\Gamma) = r^D \cdot \frac{1 - \Gamma^{-1}}{1 - \Gamma^{-M}}. \tag{13}$$

when $\Gamma \neq 1$. Otherwise, $\lim_{h \to \infty} \rho^C = r^C/M$ and $\lim_{h \to \infty} \rho^D = r^D/M$.

The high home fidelity limit reveals this nested Moran process characterised by frequency-dependent fitness at the lower level and an equivalent fixed fitness of communities at the higher level. This emerges naturally from a simple individual selection process which operates within communities and between individuals of distinct communities with the frequency of replacements coupled with how often individuals interact in the same group. We note that fixation probabilities are independent of the topology of the network, i.e. the set of edges linking the homes of different communities. The number of communities $M$, their size $Q$, and the multiplayer game played by individuals are enough to determine the evolutionary outcome of the process. The same results could be obtained from alternative movement models under the limit of isolated communities of the same size, as discussed in S1 File. Given the general nature of Eqs 12 and 13, they can be used to assess the viability of cooperation under social dilemmas in any network of communities.

Social dilemmas are characterised by the conflict between cooperation as a socially optimal strategy and defection as an individually optimal strategy [39, 51]. Given this definition, we should expect the first to excel in between-community replacements and the second at within-community fixation. The balance between these two factors is present at each step of the higher level (community) fixation process, as is represented in Fig 2. Therefore, condition $\rho^C > \rho^D$ is met in the following circumstances:

$$\frac{f_{Q,0}^C}{f_{0,Q}^D} > \left( \frac{r^D}{r^C} \right)^{1 + \frac{1}{M-1}}. \tag{14}$$

This condition is more easily met when the size of the network is increased. Under $M \to \infty$, it becomes equivalent to $\Gamma < 1$, further implying that $\rho^C > 1/N > \rho^D$ and that there is one and only one stable strategy. This shows that the definition of $\Gamma$ encapsulates the balance between the socially and individually optimal strategies, and is enough to determine the outcome of the evolutionary process under large networks.

**3.1.2 Fixation probabilities under DBB and BDD dynamics.** The DBB and BDD dynamics lead to different quantitative results as transition probability ratios in the resulting Markov chain are different from the previous four dynamics. Fixation probabilities are obtained in a parallel way to the ones presented in 12 and 13, using the following corrected values of within-community fixation probabilities $r^C$ and $r^D$, and transition probability ratios $\Gamma$:

$$r_{DBB/BDD}^C = \frac{1}{1 + \sum_{j=1}^{Q-1} \prod_{c=1}^{j} \frac{f_{c,Q-c}^D}{f_{c,Q-c}^C} \cdot \left( 1 + \frac{f_{c,Q-c}^C - f_{c,Q-c}^D}{T_{DBB/BDD}(c, Q-c) - f_{c,Q-c}^C} \right)}, \tag{15}$$

$$r_{DBB/BDD}^D = \frac{1}{1 + \sum_{j=1}^{Q-1} \prod_{d=1}^{j} \frac{f_{Q-d,d}^C}{f_{Q-d,d}^D} \cdot \left( 1 + \frac{f_{Q-d,d}^D - f_{Q-d,d}^C}{T_{DBB/BDD}(Q-d, d) - f_{Q-d,d}^D} \right)}, \tag{16}$$

$$\Gamma_{DBB/BDD} = \left( \frac{f_{0,Q}^D}{f_{Q,0}^C} \right)^2 \cdot \frac{r_{DBB/BDD}^D}{r_{DBB/BDD}^C}, \tag{17}$$

with $T_{DBB/BDD}$ denoting the total weight-fitness correction factors under those two dynamics, which are positive as evident in their definition:

$$T_{DBB}(c, d) = c \cdot f_{c,d}^C + d \cdot f_{c,d}^D, \tag{18}$$

$$T_{BDD}(c, d) = d \cdot f_{c,d}^C + c \cdot f_{c,d}^D. \tag{19}$$

There are two main distinctions between these equations and those derived in the previous section for the remaining dynamics. On one side, both DBB and BDD amplify between-community replacement events, owing to the squaring of the communal fitness ratio in 17. At the same time, they suppress within-community selection, as can be concluded from the additional coefficients multiplied by the fitness ratio in Eqs 15 and 16. The condition $\rho^C > \rho^D$ leads to

$$\frac{f_{Q,0}^C}{f_{0,Q}^D} > \left(\frac{r_{DBB/BDD}^D}{r_{DBB/BDD}^C}\right)^{\frac{1}{2}\left(1+\frac{1}{M-1}\right)}, \tag{20}$$

where the right-hand side is closer to 1 than that of Eq 14, thus benefiting cooperation.

## 3.2 Within- and between-community effects under weak selection

### 3.2.1 Fixation probabilities under weak selection.
Further considering the weak selection limit $w \to 0$, the fixation probabilities presented in section 3.1 can be expanded, leading to the following equations (see section 3 of S1 File for more details):

$$\rho^C \approx \frac{1}{MQ} + \frac{w}{2}\left[\frac{1}{Q}\left(1 - \frac{1}{M}\right)\Delta^{CD} + \left(1 + \frac{1}{M}\right)\delta^C - \left(1 - \frac{1}{M}\right)\delta^D\right], \tag{21}$$

where

$$\Delta^{CD} = R_{Q,0}^C - R_{0,Q}^D = -\Delta^{DC}, \tag{22}$$

$$\delta^C = \left.\frac{\partial r^C}{\partial w}\right|_{w \to 0} = \frac{1}{Q^2}\sum_{c=1}^{Q-1}(Q - c)\left[R_{c,Q-c}^C - R_{c,Q-c}^D\right], \tag{23}$$

$$\delta^D = \left.\frac{\partial r^D}{\partial w}\right|_{w \to 0} = \frac{1}{Q^2}\sum_{d=1}^{Q-1}(Q - d)\left[R_{Q-d,d}^D - R_{Q-d,d}^C\right]. \tag{24}$$

Eq 21 comprises three terms which are defined in Eqs 22–24. The term $\Delta^{CD}$ embodies the contribution of between-community events and corresponds to the difference between payoffs of communal cooperators and communal defectors. The sign of this term is determined by which of the two strategies is socially optimal. The terms $\delta^C$ and $\delta^D$ represent the contributions originating from the within-community fixation process of cooperators and defectors, respectively. Considering $\rho^D$ leads to the swapping of superscripts $C$ and $D$ on these three terms.

The expansion assumes a different form under the DBB and BDD dynamics, both of which result in the following equation:

$$\rho_{DBB/BDD}^C \approx \frac{1}{MQ} + \frac{w}{2}\left[2\frac{1}{Q}\left(1 - \frac{1}{M}\right)\Delta^{CD} + \left(1 - \frac{1}{Q-1}\right)\left(1 + \frac{1}{M}\right)\delta^C - \left(1 - \frac{1}{Q-1}\right)\left(1 - \frac{1}{M}\right)\delta^D\right]. \tag{25}$$

This reflects the aspects highlighted in the previous section about the impact of these dynamics. We observe the amplification of between-community selection by a factor of 2, and the suppression of within-community selection by a factor of $1 - 1/(Q - 1)$.

Each of the three contributing terms present in Eqs 21 and 25 shows a correction coefficient related to the finiteness of the network, which naturally vanishes under $M \to \infty$. Increasing the network size magnifies the relative impact of between-community replacement events on the fixation probability. At the same time, it increases the impact of the within-community fixation of residents but makes the within-community fixation of mutants relatively less significant than it is in smaller networks. In the limiting case where there are only two communities ($M = 2$), this last term exhibits a finite network correction coefficient three times larger than that of the within-community fixation of residents. This is so because the fixation of the original mutant in its community takes an increased importance in the overall process.

Increasing the size of communities decreases the impact of between-community contributions under both dynamics. Simultaneously, it amplifies the impact of within-community contributions under DBB and BDD dynamics. From Eq 25, we conclude that under the smallest communities ($Q = 2$), the expansion of fixation probabilities under DBB and BDD dynamics is reduced to a single term depending on $\Delta^{CD}$, and within-community fixation terms vanish. In a mixed subpopulation of one cooperator and one defector, both types have the same probability of being chosen first and the resulting replacement event is then certain to occur. Therefore, within-community fixation probabilities are equal to 1/2 for both types, regardless of the payoffs received by individuals. This remains true under stronger selection as was noted in [26].

**3.2.2 General social dilemmas under weak selection.** Consider the general social dilemmas defined in Table 1. We calculate the values of each of the three contributions $\Delta^{CD}$, $\delta^C$ and $\delta^D$ under all of the dilemmas introduced there, and present them in Table B of S1 File.

Under public goods dilemmas, the term $\Delta^{CD}$ is positive when cooperation is the socially optimal strategy. This happens when the reward for cooperating is sufficiently high, provided communities have a size capable of producing the reward. In the same dilemmas, the terms $\delta^C$ and $\delta^D$ exhibit negative and positive signs, respectively, due to defection being a dominant strategy.

Under the HD dilemma, the contribution $\Delta^{CD}$ remains positive regardless of reward value. The contributions $\delta^C$ and $\delta^D$ can be negative and positive for high $V/K$, positive and negative for low $V/K$, and both positive for intermediate $V/K$ when $Q > 2$. These patterns reflect that cooperation is always socially optimal in this dilemma, while within a fixed group it maintains anti-coordination properties.

We observe that cooperation can evolve under sufficiently large $V/K$ in public goods games, irrespective of the number of communities $M$, their size $Q$ (provided it allows them to produce a reward), and how they are connected. This is true even in the limiting case of two arbitrarily large communities. It is so because the contribution of between-community events can be made arbitrarily large by increasing $V$, while the remaining contributions remain constant. Although the CPD does not meet these criteria, we demonstrate this conclusion remains valid in the more detailed analysis in section 3.4. Similarly, under the HD dilemma, cooperation can evolve under sufficiently low $V/K$ irrespective of the number and size of communities, and their connections.

Moreover, based on Eqs 21 and 25 and the particular values their terms hold under each public goods dilemma, we conclude in section 4.1 of S1 File that decreasing the size of the network has a detrimental effect to cooperation under all public goods dilemmas. Smaller networks systematically lead to stricter conditions for the evolution of cooperation in public goods dilemmas. Conversely, no consistent trend emerges in the HD dilemma.

Summing the expansions obtained for the fixation probabilities of cooperators and defectors, we arrive at the following equation:

$$\rho^C + \rho^D \approx \frac{2}{MQ} + \frac{w}{M}(\delta^C + \delta^D), \tag{26}$$

where, under the DBB/BDD dynamics, an additional coefficient $1 - 1/(Q-1)$ is included in the second term on the right-hand side. It is worth noting that when the difference between the payoffs of cooperators and defectors in the same group is constant, the contributions of the within-community fixation processes of cooperators and defectors to Eqs 21 and 25 are symmetric, i.e. $\delta^C = -\delta^D$. For such dilemmas, there is always one and only one stable strategy under weak selection. This is true for all social dilemmas discussed here, except for the S and the TS with $Q > L + 1$, where bi-stability is possible ($\delta^C + \delta^D < 0$), and the HD dilemma, which allows for mutual fixation ($\delta^C + \delta^D > 0$). As established in section 3.1, under $M \to \infty$ there is one and only one stable strategy, determined by the value of $\Gamma$. This is in agreement with the fact that, for the remaining dilemmas, the second term on the right-hand side of Eq 26 vanishes under large networks. We conclude that both weak selection and a large number of communities often lead to simple dominance cases. Based on these findings, we emphasise that in all public goods dilemmas, if the fixation of cooperators is favoured under weak selection or large networks, then the fixation of defectors won't be (and vice versa). In the next section, we will extend our analysis, systematically presenting the conditions under which cooperation evolves for all social dilemmas.

## 3.3 The rules of cooperation under general multiplayer social dilemmas

In this section, we further extend our analysis of general multiplayer social dilemmas. Cooperation evolves successfully, i.e. $\rho^C > \rho^{neutral} > \rho^D$, for larger numbers of communities if

$$\Delta^{CD} > Q(\delta^D - \delta^C). \tag{27}$$

This rule is obtained considering that the first-order term of the weak selection expansion in Eq 21 has to be positive. The equation above is valid under the BDB/DBD/LB/LD dynamics, whereas for the DBB/BDD dynamics, a multiplying factor $(1/2)(1 - 1/(Q-1))$ is added to the right-hand side of the equation.

We obtain the condition under which cooperation evolves for each of the social dilemmas studied here, for all community sizes $Q$ and the six evolutionary dynamics, and present them in Table 3. The contributions $\Delta^{CD}$, $\delta^C$ and $\delta^D$ for each social dilemma are presented in Table B of S1 File. Cooperation can evolve under all of the social dilemmas approached for at least some of the explored dynamics. We opted to show the rules obtained under a high number of communities to allow a systematic analysis of the dilemmas, as obtaining them for arbitrary values of $M$ was attainable but often intricate. These limits were considered in a particular order: first $h \to \infty$, then $w \to 0$, and finally $M \to \infty$. The order of these limits is relevant, given that different orders can lead to distinct fixation probability expansions and conditions for the evolution of cooperation [50], as well as generate or erase surprising finite population effects [52]. In section 4.2 of S1 File, we analyse the validity of the simple rules presented here when these limits are relaxed.

The results presented in this table suggest that social dilemmas split into distinct groups. Non-threshold public goods dilemmas such as the PD, the VD, the S, and the PDV allow cooperators to evolve under any community size if the reward-to-cost ratio $V/K$ surpasses a critical value dependent on $Q$. This value is the same under the PD and the VD, but lower under the S and the convex PDV ($w > 1$), and higher under the concave PDV ($w < 1$). The CPD presents a

**Table 3. Rules for the evolution of multiplayer cooperation under networks of communities.** We assume a large number $M$ of communities and that they are composed of at least two individuals ($Q \geq 2$). These conditions guarantee that $\rho^C > \rho^{neutral} > \rho^D$. We denote the harmonic series as $H_Q = \sum_{i=1}^{Q} i^{-1}$. Under $Q = 1$, the derived conditions are the following: cooperation never evolves under the CPD, TVD, SH, FSH, and TS (assuming that $L \geq 2$), cooperation evolves for $V/K > 1$ under the PD, PDV, VD and S, and both strategies are neutral under the HD. These results are valid under arbitrary values of $w$ and $M$, and they are the same under all six dynamics.

| Multiplayer Game | Evolution of cooperation under cooperat | |
|---|---|---|
| | BDB/DBD/LB/LD | DBB/BDD |
| CPD | $\emptyset$ | $V/K > (Q-1)$ |
| PD, VD | $V/K > Q$ | $V/K > \dfrac{Q}{2}$ |
| PDV | $V/K > \dfrac{1-\omega}{1-\omega^Q} Q^2$ | $V/K > \dfrac{1-\omega}{1-\omega^Q} \dfrac{Q^2}{2}$ |
| S | $V/K > H_Q$ | $V/K > \left( H_Q - \dfrac{1}{2} \dfrac{Q}{Q-1} H_{Q-1} \right)$ |
| TVD, SH | $\begin{cases} V/K > Q & Q \geq L \\ \emptyset & Q < L \end{cases}$ | $\begin{cases} V/K > \dfrac{Q}{2} & Q \geq L \\ \emptyset & Q < L \end{cases}$ |
| FSH | $\begin{cases} V/K > Q^2 & Q \geq L \\ \emptyset & Q < L \end{cases}$ | $\begin{cases} V/K > \dfrac{Q^2}{2} & Q \geq L \\ \emptyset & Q < L \end{cases}$ |
| TS | $\begin{cases} V/K > H_Q - H_L + 1 & Q \geq L \\ \emptyset & Q < L \end{cases}$ | $\begin{cases} V/K > 1/2 \left[ 1 + \left( 1 - \dfrac{1}{Q-1} \right)(H_Q - H_L) \right] & Q \geq L \\ \emptyset & Q < L \end{cases}$ |
| HD | $V/K < \dfrac{Q - 1/Q - H_{Q-1}}{H_{Q-1}}$ | $V/K < \dfrac{\dfrac{Q-1}{Q-2}(Q - 2/Q) - H_{Q-1}}{H_{Q-1}}$ |

distinct landscape, where cooperation only evolves under the DBB and BDD dynamics. The critical value of the reward-to-cost ratio in this dilemma is the lowest of all non-threshold public goods games. We will analyse this dilemma in the following section.

Threshold dilemmas such as the TVD, the SH, the FSH, and the TS have a critical value of the reward-to-cost ratio, above which cooperation evolves, only if the size of communities is at least of the same size as the public goods production threshold ($Q \geq L$). Otherwise, cooperation can never evolve regardless of the value of $V/K$. The TVD and the SH lead to the same conditions, which coincide with the PD and the VD when $Q \geq L$. The TS leads to lower critical values of the reward-to-cost ratio, and the FSH leads to higher values. We further note that the critical values obtained under the FSH when $Q \geq L$ are simply the ones obtained under the PDV with $\omega \to 0$. Critical values under threshold games generally don't depend on $L$, although their existence does. The exception to this is the TS dilemma, under which a larger production threshold decreases the critical value of the reward-to-cost ratio when communities are large enough to produce rewards.

The HD dilemma, which unlike the others is a commons dilemma, behaves distinctively from all of the remaining dilemmas. The reward-to-cost ratio has to be lower than a critical value for cooperation to evolve. It is clear that in this case, high rewards are detrimental to the evolution of cooperation.

We note that the critical value of the reward-to-cost ratio under public goods dilemmas always increases with the size of communities and regardless of the used evolutionary dynamics. This allows us to provide the visual representation from Fig 3 with the areas under which cooperation evolves for community sizes up to a given value. Additionally, as mentioned in section 3.2, considering lower values of $M$ always leads to stricter conditions for the evolution of cooperation. This reinforces the conclusion that populations organised into large networks

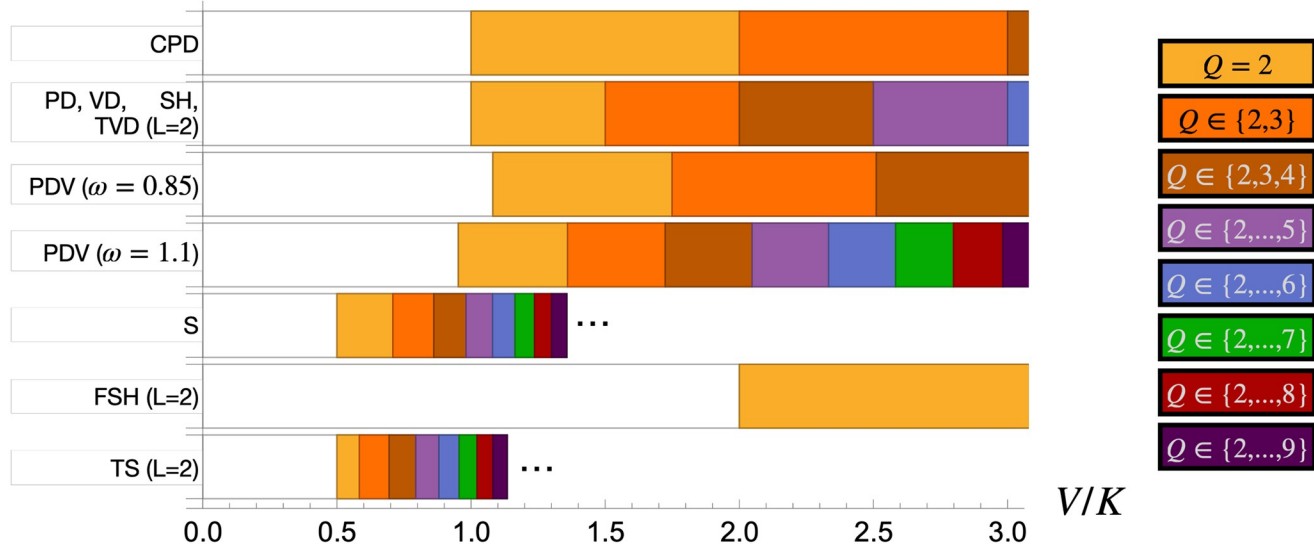

**Fig 3. Regions under which cooperation evolves for each public goods dilemma.** Each coloured region covers the values of the reward-to-cost ratio, i.e. $V/K$, under which cooperation evolves for a given set of community size values which are stated in the legend. These regions are obtained from the rules for the evolution of cooperation presented in Table 3. Under low enough values of $V/K$, all dilemmas have uncoloured regions, as no community size allows the evolution of cooperation. We opted for not showing the areas of the S and the TSD dilemmas with higher values of $V/K$ (starting from the ellipsis), as coloured regions quickly decreased in size: at $V/K = 3$, cooperation evolves for any $Q \leq 231$ under the S and $Q \leq 377$ under the TS when $L = 2$.

of small communities lead to a larger region of the parameter space under which cooperation evolves. This is so because cooperators hold an advantage in between-community reproduction events (intensified under large $M$), but they are disadvantaged in within-community fixation processes (minimised under small $Q$).

In this context, the HD dilemma has key differences compared with the public goods dilemmas. Under this game, cooperators hold an advantage in between-community reproduction events for any payoff parameters. Regarding within-community fixation processes, defectors hold an advantage in small communities, but cooperators are the ones doing so in larger communities. However, there is a second overlapping effect which is described in section 3.2: increasing the size of communities decreases the impact of between-community reproduction and increases the impact of within-community fixation. Under the BDB dynamics, the second effect is not strong enough and the first effect dominates: communities with larger size always lead to higher critical values below which cooperators fixate, therefore benefiting them. However, under the DBB/BDD dynamics, both effects interplay and each dominates at a different scale of community sizes. Cooperators always evolve when $Q = 2$ because fixation depends only on between-community reproduction. When increasing the community size to $Q = 3, 4$, the emerging critical values below which cooperation evolves decrease with community size because of the increased importance of within-community fixation beneficial to defectors in those community sizes. However, for larger values of $Q \geq 5$, cooperation evolves for larger regions of $V/K$ when increasing $Q$ because within-community fixation starts benefiting cooperators.

Comparing the critical values obtained between the different evolutionary dynamics, we note that the DBB and BDD dynamics always extend the values of $V/K$ for which cooperation successfully fixates when compared to the remaining dynamics. They therefore have lower critical values in all public goods dilemmas and higher critical values in the HD dilemma. We note the extreme case of the CPD, under which cooperators never evolve under the BDB and

equivalent dynamics, but find an evolutionary way under the DBB and BDD dynamics. These results can be explained by the fact that these dynamics when compared to the remaining, amplify the impact of between-community replacement terms (where cooperators succeed relative to defectors), and suppress within-community selection terms (where defectors succeed).

### 3.4 The Charitable Prisoner's Dilemma and pairwise cooperation

The CPD is a particular game of interest among public goods dilemmas. Under the CPD, cooperators do not benefit from their own contributions to public goods. This assures not only that individuals have equal gains from switching, but also that the gains are the same for all group sizes. In other words, the cost $K$ is the effective cost that a cooperator pays for not defecting, regardless of group composition and size. This game is thus a social dilemma regardless of how large the reward is and the size of the interacting group [39]. Other games have equal gains from switching, but the gains vary with group size. One such game is the PD, which was introduced in [41], under which the cost of cooperating is $K - V/Q$, and therefore may not even present a social dilemma under some payoff choices and group sizes [39].

Table 3 shows that cooperation evolves in the CPD when $V/K > (Q - 1)$. Given our particular interest in it, we present here the condition for the evolution of cooperation obtained under the CPD when a finite number of communities $M$ is considered:

$$V/K > (Q - 1) \cdot \frac{1 - \dfrac{2}{MQ}}{1 - \dfrac{2(Q-1)}{MQ}}. \tag{28}$$

This rule quantifies the detrimental effect that considering a lower number of larger communities (lower $M$ and higher $Q$) has on the evolution of cooperation. At the same time, it materialises a fundamental result: cooperation can evolve provided rewards are high enough, for any given community size and number, and regardless of the connections between them. This is a remarkably general result that works for the smallest networks of two communities under which cooperation evolves if $V/K > (Q - 1)^2$.

A parallel result was attained in [25] by considering the pairwise donation game in an evolutionary graph which is split into $M$ cliques of $Q$ individuals each. Individuals within the same clique are considered to have unit-weighted edges and there is an arbitrary set of infinitesimal edges between individuals of different cliques. The vanishing edges act to isolate the individuals within each clique, guaranteeing that cooperation can always evolve in the pairwise donation game if $V/K$ is high enough.

The rules obtained under the CPD are parallel not only to the clique structures explored in [25] but also to the results obtained in [22] for large regular networks. They showed that cooperation can evolve under the DBB dynamics if the reward-to-cost ratio is larger than the average number of neighbours each individual has on an interaction network. We note that in our model and the particular limit of large home fidelity, each individual regularly interacts with $Q - 1$ others and that this is exactly the critical value of the reward-to-cost ratio under the DBB dynamics. However, the results obtained here for networked communities allow cooperation to evolve under the smallest networks when the corrected rule presented in Eq 28 is met, thus going beyond the large network assumption.

At the same time, when interacting via the CPD, cooperators can never evolve if the evolutionary dynamics considered are the BDB/DBD/LB/LD dynamics, as shown in section 2.2 of S1 File for arbitrary values of intensity of selection and number of communities. This had been already hypothesised in [26] for the general formulation of the territorial raider movement

model, similar to what was observed in previous evolutionary graph theory models [22]. However, we note that this feature of the BDB dynamics is a singular case when stochastic combinations of different types of dynamics are considered, as was shown in [27].

The CPD can be seen as a multiplayer extension of the pairwise donation game and as such, the two games may lead to analogous results. More generally, the exploration of higher-order interactions leads to different interacting structures and evolutionary outcomes [28], even in other cases where the multiplayer game considered is a natural extension of its pairwise version. However, in the particular limit studied here, individuals always interact within their own communities which are all of the same size. Therefore, the average payoffs obtained in a well-mixed community playing the pairwise donation game are the same as the payoffs obtained in a group of fixed size repeatedly playing the CPD. This is no longer the case when lower home fidelity values are considered, and new higher-order differences are expected to arise in that context.

## 4 Discussion

In the present work, we use the territorial raider model previously approached in [26, 36–38], a fully independent movement model which is described by one single parameter, the home fidelity of individuals. The general framework originally proposed in [29] can be thought of as a natural extension of evolutionary graph theory to multiplayer interactions, under which replacement events between individuals in the population occur proportionally to how often they interact. We focus on the limit of high home fidelity, under which individuals interact mostly within their community with the rare occurrence of cross-community group interactions. We derive the evolutionary dynamics in this limit, which is revealed to be a nested Moran process resembling metapopulation models where migration is coupled with selection (these are classified in [15]), but is asymptotically rare as is considered in [20]. Therefore, we show that metapopulation dynamics of multiplayer interactions can be derived from basic evolutionary graph theory assumptions. This derivation is achieved without considering between-community events to be of a different nature through the introduction of migration [15, 20], group splitting and replacement [16–18], or two or more levels of intensity of selection [19, 20]. The same results could be obtained from alternative movement models under the limit of isolated communities of the same size, as discussed in S1 File.

In this context, we show that whether a strategy evolves or not depends on the advantage it holds against other strategies in two contexts: when in homogeneous groups and when in within-community fixation processes. Multiplayer social dilemmas involve the existence of a conflict between cooperating as a socially optimal strategy and defecting as an individually optimal strategy [39, 51]. Therefore, we obtain a general condition for the evolution of cooperation which translates into a simple balance between its advantages in homogeneous communities and its disadvantages over within-community fixation processes.

Applying this balance to the multiplayer social dilemmas explored in [39], we obtain simple rules for the evolution of multiplayer cooperation in community-structured populations. These depend on the reward-to-cost ratio, and the number and size of communities. Cooperation evolves under all social dilemmas for any given number of communities, as long as there are at least two, that they are large enough to produce rewards (when applicable), and that the rewards are high enough in public goods dilemmas or low enough in the HD dilemma (a commons dilemma focused on the fair consumption of preexisting resources). In public goods dilemmas, cooperation evolves more easily when the costs of production are shared (the S and the TS dilemmas—see [24, 42] for an account of this), when the reward production function is supralinear (the PDV), and when individuals benefit from their own production (all public

goods dilemmas, except for the CPD). However, finding that cooperation can evolve under the CPD in any community-structured population was remarkable by itself, given that this dilemma does not have any of the above characteristics and extends some of the strictest properties of the pairwise donation game to larger group sizes. Other characteristics of public goods dilemmas could be assessed in the future by considering asymmetric reward contributions and productivities (quantified as each individual's reward-to-cost ratio) [53], or even different mobility distributions and costs [35].

Moreover, the general results derived are not restricted to public goods dilemmas. The multiplayer HD game revealed an entirely different landscape when compared to its pairwise equivalent, the S dilemma. The differences between the two types of multiplayer dilemmas highlight that the considerations taken when extending pairwise games to higher-order interactions may reveal fundamental differences between them. These differences materialise here in the distinction between dilemmas focused on production (public goods dilemmas) and fair consumption of a preexisting resource (commons dilemmas). Furthermore, the use of the general rule obtained for the evolution of cooperation can be extended to the study of systems where evolutionary games have been employed, such as in AI monitoring [54], disease evolution and spread [55], environmental governance [56], and healthcare investment [57].

Remarkably, the derived dynamics did not depend on how communities were connected, with the community effects overwriting other potentially overlapping structural effects. It was observed in [36] that high home fidelity led to a simple fixed fitness Moran process independent of topology in the territorial raider model with $Q = 1$, which is simply a particular case of the more general nested Moran process we derived in this work. For general home fidelity values, it was shown in [37, 38] that temperature and average group size can be good predictors of fixation probabilities in the HD dilemma and the CPD, for a wide selection of topologies. Interpreted in that light, our results show that when strict subpopulation temperature as defined in [26] is zero and the size of the network of places is fixed, the success of the fixation process is determined by the size of communities and independent of other topological features. This is in contrast with the models under which network topology plays a key role, such as evolutionary games on static pairwise graphs [22, 23, 25] and satisfaction-dependent movement models [33, 34].

Public goods dilemmas consistently lead to the evolution of cooperation down to lower values of the reward-to-cost ratio when a larger number of smaller communities is considered. This is in line with what is observed in alternative community and deme models [20, 26, 58], and multilevel selection models [16–18]. The only exception to this is presented by multilevel public goods games when punishment is introduced, in which case larger communities are beneficial for cooperation [19]. It was shown in [25] that networks of isolated clusters interacting via the pairwise donation game also favour cooperation more frequently under smaller clusters and larger networks. Furthermore, strong isolated pairs were shown to be a strong predictor of cooperation in any evolutionary graphs [25]. Therefore, fragmentation into smaller social communities or groups might be one of several key mechanisms at the origin of cooperative behaviour observed around us. This is further supported by experimental studies which show that, in smaller groups, altruistic interventions occur more often [59], and free riding is less common [60]. Perhaps this helps explain why interactions in smaller groups, particularly in groups of two individuals, are consistently observed to be more prevalent in a wide range of human social interactions [6].

The results presented in this paper were obtained within the limit of high home fidelity, under which communities become asymptotically bounded interacting groups. A relaxation of this limit is expected to lead to several key differences. Firstly, we would expect an increase in the rate at which between-community events happen, tied to the occurrence of group

interactions between individuals of different communities, and therefore to the blurring of the interacting boundaries between them. In the pairwise donation game, considering less isolated clusters leads to stricter conditions for the evolution of cooperation [25]. Even though a similar trend has been observed in the CPD in some small networks [26, 36], this should not be extrapolated to larger networks and all topologies as interacting groups have variable size and the dilemma no longer has an equivalent pairwise representation. In that case, the group structure underlying the multiplayer interactions depends not only on the size and number of communities but also on how the home nodes of each community are connected. Accounting for interacting groups in a different way may therefore lead to fundamentally different results, even when the underlying social structure remains very similar or the same, as was previously reported in [61, 62]. Parallel approaches to higher-order interactions show surprisingly high cooperative states under a class of multiplayer extensions of the prisoner's dilemma [63]. Similar effects may emerge under communities with blurred boundaries, namely when considering dilemmas with non-rivalrous public goods and/or shared costs, such as the S dilemma, given their propensity to evolve cooperation under high group size variance [24, 42, 62]. The framework used in this work shows its flexibility once again, leading to evolutionary dynamics similar to metapopulation and deme models under large home fidelity, while offering the possibility to explore new complex group interaction dynamics outside of that limit.

## Supporting information

**S1 File. Supplementary material.**
(PDF)

## Author Contributions

**Conceptualization:** Diogo L. Pires, Mark Broom.

**Formal analysis:** Diogo L. Pires, Mark Broom.

**Funding acquisition:** Mark Broom.

**Supervision:** Mark Broom.

**Visualization:** Diogo L. Pires.

**Writing – original draft:** Diogo L. Pires.

**Writing – review & editing:** Diogo L. Pires, Mark Broom.

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
