## [Decision Letter · Decision Letter 0]

5 Jul 2024

Dear Mr Pires,

Thank you very much for submitting your manuscript "The rules of multiplayer cooperation in networks of communities" for consideration at PLOS Computational Biology. As with all papers reviewed by the journal, your manuscript was reviewed by members of the editorial board and by several independent reviewers. The reviewers appreciated the attention to an important topic. Based on the reviews, we are likely to accept this manuscript for publication, providing that you modify the manuscript according to the review recommendations.

Please follow the suggestions of the two referres!

Sincerely,

Attila Csikász-Nagy

Academic Editor

PLOS Computational Biology

Zhaolei Zhang

Section Editor

PLOS Computational Biology

Reviewer's Responses to Questions

**Comments to the Authors:**

Reviewer #1: Authors study evolution dynamics of cooperation in multi-player interactions in mobile structured meta-populations, using methods from evolutionary game theory. They derive several general, analytical results for the limit of high home fidelity, where individuals interact mainly within their own communities.

They then applied their results to a range of social/cooperation dilemma games and considered several reproduction/behaviour update rules.

I find the paper well-written, the analyses thorough and done in a highly competent manner, and the findings important and impactful for the understanding of human cooperation in more realistic complex network settings. The literature of the evolution of cooperation is extensive but there is a lack of methodologies for analysing more realistic settings such as the ones studied in the current paper (i.e. structured meta-population with mobility). I believe the current work would make a significant contribution to this literature. The area of research is also strongly aligned with PLOS Computational Biology.

Therefore, I would be happy to recommend publication of the paper in the present form. If there is a chance for revision, authors might consider the following optional suggestions:

1) The paper includes a large number of parameters — it might be useful to include a table (in the main text of Supporting Information) that summarises these parameters, including relevant information such as their ranges, etc.

2) The paper focuses on cooperation/social dilemma settings. It seems meta-population evolutionary game methods have been applied to several other contexts, e.g. AI governance and safety (see e.g. "Trust AI Regulation? Discerning users are vital to build trust and effective AI regulation." arXiv preprint arXiv:2403.09510 (2024), and "Both eyes open: Vigilant Incentives help Auditors improve AI Safety." Journal of Physics: Complexity (2024).), environmental monitoring (”Paradigm shifts and the interplay between state, business and civil sectors." Royal Society open science 3.12 (2016): 160753.) and healthcare investment ("Toward Understanding the Interplay between Public and Private Healthcare Providers and Patients: An Agent-based Simulation Approach." EAI Endorsed Transactions on Industrial Networks and Intelligent Systems 7.24 (2020): 166668.).

I think the developed methods in the current paper can be very useful to study behavioural evolution in these emerging contexts.

Reviewer #2: The authors examine the spread of cooperation in the network of communities. The communities consist of Q individuals and there are M of them arranged in a graph.

With high probability, the individual interacts with his own community, however, with small probability (proportional to a parameter h) it might interact with other neighboring communities.

Everyone plays some game (such as a prisoners' dilemma or other dilemmas) and obtains a payoff.

The payoff influences the fitness of the individual. Then, one step of the Moran process (death-birth or birth-death) is performed.

The authors compare the setting with the neutral case, where the payoff does not influence the fitness and one random individual fixates with probability 1/MQ.

The paper shows the range of parameters where the PD (or any dilemma) leads to fixation probability of cooperation is above 1/MQ for h in the limit.

The methods are technically challenging, but not surprising.

The most important ingredient of the proof is to observe that the probability of cooperators conquering a community times the probability of spreading (which depends on fitness) is higher than the probability of defectors conquering a community and spreading.

This result is obtained because of the "high home fidelity", i.e. the probability that one type fixates in the community happens before any between-community event.

The authors show the results for 10 different dilemmas, six different update rules, and weak and strong selections.

This is impressive, however the techniques to obtain the results are very similar.

----------

The manuscript has lot of strengths and studies an important topic, however, some changes are needed before I can recommend an acceptance.

The model itself is fragile and the weaknesses and border cases are not properly addressed.

Namely: the probability that an individual interacts with some other group is 1/(h+d) where d is the degree.

This choice seems robust, but changing the probability to 1/h*1/d (with probability 1/h it "misses" his group and then chooses randomly) gives a very different dynamic.

(more similar to the moran process).

Therefore some strong justification is needed and some clarification about the limits of the model will also be helpful.

A clearer explanation for Figure 1 is desirable: it seems that in the next step, the individual can move and then stay in a different community.

(It's not true, but from the first reading it seems correct and this process seems more interesting.)

The fitness of an individual (eq 3) makes me a bit uncomfortable.

The model should simulate a discrete process. That means the payoff should be the realization of the interaction within the community (with current members), not the expectation over all possible configurations.

Here, it might be desirable that everyone having the payoff computed from the original community.

(The current choice looks more general, but it's not)

Again, in eq (5), there could be more straightforward (with weight 1/(h+d) we spread into different communities).

This again looks more general than it is and makes the model unnecessarily complicated.

Later, some computations might be moved to SI, it would ease the reading, but I'm fine either way.

**Have the authors made all data and (if applicable) computational code underlying the findings in their manuscript fully available?**

Reviewer #1: None

Reviewer #2: Yes

PLOS authors have the option to publish the peer review history of their article (what does this mean?). If published, this will include your full peer review and any attached files.

Reviewer #1: No

Reviewer #2: No

Figure Files:

Data Requirements:

Reproducibility:

References:

---

## [Editor Report · Decision Letter 1]

5 Aug 2024

Dear Mr Pires,

We are pleased to inform you that your manuscript 'The rules of multiplayer cooperation in networks of communities' has been provisionally accepted for publication in PLOS Computational Biology.

Best regards,

Attila Csikász-Nagy

Academic Editor

PLOS Computational Biology

Zhaolei Zhang

Section Editor

PLOS Computational Biology

---

## [Editor Report · Acceptance letter]

13 Aug 2024

PCOMPBIOL-D-24-00868R1 

The rules of multiplayer cooperation in networks of communities

Dear Dr Pires,

I am pleased to inform you that your manuscript has been formally accepted for publication in PLOS Computational Biology. Your manuscript is now with our production department and you will be notified of the publication date in due course.

With kind regards,

Anita Estes
